# Daily exposure to stressors, daily perceived severity of stress, and mortality risk among US adults

**Dana A. Glei**◉*, **Maxine Weinstein**

Center for Population and Health, Georgetown University, Washington, District of Columbia, United States of America

* dag77@georgetown.edu

**Data Availability Statement:** The MIDUS data are publicly available from ICPSR at https://www.icpsr.umich.edu/web/ICPSR/series/203/. In this analysis, we use the data from Waves 1-3 of the main survey (https://doi.org/10.3886/ICPSR02760.v19;

## Abstract

Prior studies of perceived stress and mortality have yielded mixed results, but most are based on one-time measurements of perceived stress. We use daily diary data from the Midlife in the United States study to measure exposure to stressors and perceived severity of stress and investigate their associations with mortality. We also explore whether the associations vary by age and assess whether the associations are stronger for extrinsic than intrinsic mortality, which is more likely to be aging-related. The analysis included 4,756 observations for 2,915 respondents aged 21–95 who participated in at least one of three waves (1996–97, 2004–09, 2017–19) of the National Study of Daily Experiences. Participants reported daily stressors and perceived severity on 8 consecutive evenings at each wave. Mortality was followed through December 31, 2021. In fully-adjusted models, daily exposure to stressors was associated with mortality, but only at younger ages (HR = 1.20 per SD at age 50, 95% CI: 1.01–1.42). The association was slightly stronger for extrinsic (HR = 1.31 per SD at age 50, 95% CI: 1.01–1.69) than for intrinsic mortality, which was not significant (HR = 1.24 per SD at age 50, 95% CI: 0.98–1.56). When we used an alternative measure of daily perceived severity of stress, the demographic-adjusted association appeared to be similar in magnitude, but after careful adjustment for potential confounding with health status, the association weakened and was no longer statistically significant (HR = 1.17 per SD at age 50, 95% CI: 0.99–1.37). Perceived severity was not significantly associated with either extrinsic or intrinsic mortality even at age 50. Most Americans die at older ages, where stress exposure does not appear to be significantly associated with mortality. Nonetheless, our results suggest that stress exposure is more strongly associated with mid-life mortality, which has an undue influence on overall life expectancy.

## Introduction

The links among stress, its perception, and well-being have been proposed for close to two millennia. Marcus Aurelius (121–180 AD) commented [1], "If you are distressed by anything external, the pain is not due to the thing itself, but to your estimate of it; and this you have the power to revoke at any moment." Similarly, Epictetus (c. 50–135 AD) noted [2], "It is not events that disturb people, it is their judgments concerning them." And more than a

https://doi.org/10.3886/ICPSR04652.v8; https://doi.org/10.3886/ICPSR36346.v7), Waves 2 and 3 of the Milwaukee cohort (https://doi.org/10.3886/ICPSR22840.v6; https://doi.org/10.3886/ICPSR37120.v4), Waves 1-3 of the NSDE (https://doi.org/10.3886/ICPSR03725.v6; https://doi.org/10.3886/ICPSR26841.v2; https://doi.org/10.3886/ICPSR38529.v1), and the mortality follow-up data (https://doi.org/10.3886/ICPSR37237.v4).

**Funding:** This work was supported by the National Institute on Aging [https://www.nia.nih.gov/, grant numbers P01 AG020166, U19AG051426, and U19 AG051426-06A1 to Carol Ryff (PI of the MIDUS project)] and the Graduate School of Arts and Sciences, Georgetown University (https://grad.georgetown.edu/). The funders had no role in study design, data collection and analysis, decision to publish, or preparation of the manuscript.

**Competing interests:** The authors have declared that no competing interests exist.

millennium later, Shakespeare wrote in Hamlet [3, Act II, Scene 2]: "There is nothing either good nor bad but thinking makes it so."

Everyone is exposed to potential stressors, but people differ in the extent to which they interpret a situation as stressful and how it affects their health and mortality. An individual's subjective appraisal (i.e., perception) is crucial to understanding the link between exposure to stressors and the physiological and behavioral response [4, 5]. One advantage of measuring perceived stress is that it explicitly incorporates the respondent's appraisal (i.e., the extent to which the person feels "stressed"). A disadvantage is that poor health could be a confounder that influences both the perception of stress and mortality; that is, the apparent association could be at least partly spurious. Thus, if we want to evaluate the relationship between perceived stress and mortality, it is important to control for underlying health as a potential confounder.

Prior studies of the association between perceived stress and mortality have yielded mixed results. Some studies have reported an association between perceived stress and mortality [6–9], but one of those studies found the association was no longer significant after adjusting for health conditions [9]. Others found no association [10, 11] or reported an association among men but not women [12]. Most studies are based on one-time measurements of perceived stress [6–8, 11, 12]. It is rare to have daily measures of perceived stress over more than one week. One unique study used daily diary data to evaluate stress exposure over an 8-day period [13]. Although they did not measure perceived stress per se, they found a positive association between number of daily stressors and mortality. We build on this study by incorporating perceived severity of daily stress.

We expect the association with perceived stress to be stronger for some causes of death than others. An earlier study in Denmark found the association with perceived stress was stronger for mortality from external causes than for all-cause mortality, at least among men [12]. Extrinsic mortality generally refers to deaths resulting from external factors (e.g., environmental hazards), whereas intrinsic mortality is assumed to be a result of internal factors (e.g., aging). We suspect that extrinsic mortality is more likely to be determined by individual behavior (e.g., smoking, drug abuse, and other risk-taking behaviors) than intrinsic mortality, which is more likely to be aging-related. Daily perceived stress is likely to affect the propensity to engage in risky health behaviors. Therefore, we anticipate that daily perceived stress will be more strongly associated with extrinsic mortality than intrinsic mortality.

Prior work also suggests that association with perceived stress may vary by age. That same Danish study reported that the association was stronger at younger ages, at least among men [12]. Most deaths occur at older ages, where aging-related disease dominates, but at younger ages, individual-level behaviors are likely play a larger role, and as noted above, perceived stress may influence those behaviors. Therefore, we test whether daily perceived stress is more strongly associated with mortality at younger than at older ages.

Here, we use daily diary data from the Midlife in the United States (MIDUS) study to measure exposure to stressors and perceived severity of stress and investigate their associations with mortality. In addition, we explore whether the associations vary by age and assess whether the associations are stronger for extrinsic than intrinsic mortality.

## Methods

### Data

We used cohort data from the Waves 1–3 of the MIDUS main survey and National Study of Daily Experiences (NSDE) sub-study, which included US adults aged 21–95 at the NSDE (Table 1). Details about the sampling strategy are provided in S1 Text. At Wave 1, 1,499

**Table 1. Descriptive statistics for covariates by survey wave.**

|  | Wave 1 | Wave 2 | Wave 3 |
|---|---|---|---|
| Age at the NSDE (21–95), mean (SD)[a] | 47.7 (12.8) | 58.2 (12.1) | 67.1 (10.4) |
| Male,[b] No. (%) | 697 (46.4) | 865 (42.8) | 529 (42.8) |
| Race[b] |  |  |  |
| White, No. (%) | 1364 (91.0) | 1723 (85.2) | 1048 (84.9) |
| Black, No. (%) | 88 (5.9) | 243 (12.0) | 149 (12.1) |
| Other race, No. (%) | 47 (3.1) | 56 (2.8) | 38 (3.1) |
| Married/partnered, No. (%) | 1086 (72.4) | 1452 (71.8) | 842 (68.2) |
| Relative SES[c] (0–1), mean (SD) | 0.5 (0.3) | 0.5 (0.3) | 0.5 (0.3) |
| Ever had cancer, No. (%) | 103 (6.9) | 279 (13.8) | 241 (19.5) |
| Ever had heart trouble, No. (%) | 177 (11.8) | 352 (17.4) | 261 (21.1) |
| Stroke in the past 12 months, No. (%) | 12 (0.8) | 25 (1.2) | 16 (1.3) |
| Diabetes in the past 12 months, No. (%) | 60 (4.0) | 215 (10.7) | 167 (13.5) |
| Lung problems in past 12 months, No. (%) | 211 (14.1) | 270 (13.4) | 169 (13.7) |
| Physical limitations (-0.7–3.2), mean (SD) | 0.7 (1.3) | 1.2 (1.3) | 1.3 (1.3) |
| Daily number of stressors (0–5), mean (SD) | 0.5 (0.5) | 0.5 (0.5) | 0.5 (0.5) |
| Daily perceived severity of stress (0–14), mean (SD) | 1.0 (1.0) | 1.0 (1.0) | 0.9 (1.0) |
| Number of respondents | 1,499 | 2,022 | 1,235 |

Percentages may not add to 100% because of rounding errors.

Abbreviation: SD, Standard Deviation; SES, Socioeconomic Status.

[a] Most (>99%) of the analytic sample were aged 26–75 at NSDE1 (with only N = 1 aged 21 and N = 2 aged 76), aged 35–85 at NSDE2 (with only N = 2 aged 86), and aged 47–93 at NSDE3 (with only N = 3 aged 94–95).

[b] Measured only at baseline (Wave 1 for the main cohort, Wave 2 for the Milwaukee sample).

[c] Within each survey wave, the SES index was converted to percentile rank and rescaled to range from 0 (bottom percentile) to 1 (top percentile).

respondents participated in the NSDE (fielded March 1996-Apr 1997). At Wave 2 of the NSDE (fielded July 2004-April 2009), 793 of the respondents from NSDE Wave 1 participated again and another 1,229 new respondents completed the Wave 2 NSDE (including 180 respondents from the Milwaukee cohort). At Wave 3 of the NSDE (fielded March 2017-September 2019), 1,048 respondents were re-interviewed and there were 187 new participants.

Our analysis included 4,756 observations for 2,915 respondents who participated in at least one wave of the NSDE (N = 1,499 from Wave 1, N = 2,022 at Wave 2, and N = 1,235 from Wave 3). Of those, 412 respondents completed all three waves of the NSDE. For each survey interval, respondents were included from the date they were interviewed until the date of the next wave or their death, whichever came first; survivors who did not participate in the subsequent wave were censored at the start of fieldwork for that wave.

The MIDUS study was approved by the Educational and Social/Behavioral Science institutional review board at the University of Wisconsin, Madison [#SE-2011-0350]. Written informed consent was obtained from all participants. The MIDUS data were accessed for research purposes on 7/31/2023. The authors did not have access to information that could identify individual participants during or after data collection.

## Measures

**Mortality.** Vital status was ascertained through searches of the National Death Index, survey fieldwork, and longitudinal sample maintenance [14]. To ensure the completeness of

mortality follow-up, we analyzed deaths only through December 31, 2021 (see S2 Text for details). Among the analytic sample, there were 566 deaths.

Given the number of deaths among our analytic sample, we have limited statistical power to model cause-specific mortality. Nonetheless, we estimated auxiliary models for extrinsic (141 deaths) vs. other mortality (410 deaths). The cause of death was unknown for 15 decedents, who were excluded from the cause-specific analyses.

Extrinsic mortality was defined based on Masters et al. [15] with some modifications (see S2 Text for more details). More than half of these deaths were probably related to smoking (e.g., 38% were from respiratory diseases, mostly COPD, but there were also some deaths from emphysema and pneumonia; another 26% were from lung cancer). The next biggest share resulted from external causes (20%) and other deaths of despair (5%). COVID-19 (1%) and other infectious diseases (7%) accounted for most of the remainder.

We refer to the residual category as "intrinsic mortality," which encompassed all other deaths not defined as "extrinsic." Researchers use the term "intrinsic mortality" to refer to deaths believed to result from aging, whereas "extrinsic mortality" denotes deaths thought to stem from environmental hazards, the risk of which are more constant across age [16]. In our sample, intrinsic deaths comprised mostly deaths from several leading causes of death: heart disease (33%), cancers other than lung, bronchus, trachea, and cervix (29%), cerebrovascular disease (8%), and Alzheimer's/dementia (7%). Another 6% resulted from endocrine/nutritional/metabolic diseases (mostly diabetes). The remaining 16% came from various other causes (i.e., 6% other nervous system; 4% other digestive; 2% genitourinary; 2% other circulatory; <0.5% ill-defined; 1.5% all else).

**Daily exposure to stressors.** In the NSDE for each survey wave, daily stressors were assessed using the Daily Inventory of Stressful Events [17]. On each of 8 consecutive evenings, the respondent was asked about 7 potential stressors (responses were binary: yes/no) that may have occurred "since this time yesterday":

1. "Did you have an argument or disagreement with anyone?";

2. "Did anything happened that you COULD have argued about but you decided to LET PASS in order to AVOID an argument?";

3. "Did anything happen at work or school that most people would consider stressful?";

4. "Did anything happen at home that most people would consider stressful?";

5. "Many people experience discrimination on the basis of such things as race, sex, or age. Did anything like this happen to you?";

6. "Did anything happen to a close friend or relative that turned out to be stressful for YOU";

7. "Did anything ELSE happen to you that most people would consider stressful?"

For each of the 7 stressors, we computed the proportion of observed days (within the survey wave) in which the respondent reported exposure to that stressor. Then, we summed across stressors to obtain the per day average number of stressors within-subject at each survey wave (observed range: 0–5).

**Daily perceived severity of stress.** For each stressor reported in the NSDE, the respondent was asked "how stressful was this for you–very, somewhat, not very, or not at all?" We recoded the four response categories to range from 0 ("not at all") to 3 ("very"). If the respondent did not experience a given stressor on that day, we recoded the variable to 0. For each of the 7 stressors, we computed the mean severity across observed days (within the survey wave). Then, we summed the per day severity across the 7 stressors to obtain a within-subject

measure of daily cumulative severity at each survey wave. This measure is cumulative across the various stressors (i.e., if the respondent reported severe stress for multiple stressors, that person would have a higher score than someone with severe stress on only one stressor), but represents the per day average severity (i.e., we do not accumulate across days). Theoretically, the daily severity of stress score ranges from 0 to 21 (if the respondent reported all 7 stressors were "very" stressful on all observed days), but the maximum score was 14 (e.g., "somewhat" stressful for all 7 stressors across all observed days).

**Potential confounders.** We controlled for the following variables that are known to affect mortality rates and may also influence the perception of stress: age, sex, race, marital status (married/partnered vs. all else), socioeconomic status (SES), selected chronic conditions, and physical limitations. All confounders were treated as time-varying covariates except sex and race. Age was measured at the first day of the NSDE for each wave. Data for the other confounders come from the main survey.

Race was based on self-identification ("*What race do you consider yourself to be*?"). We retained the first two response categories (i.e., Black and/or African American; White), but because of small sample size, we combined the remaining categories (i.e., Asian or Pacific Islander; multiracial; Native American or Aleutian Islander/Eskimo; other) into a group labeled "other race." We did not include Hispanic ethnicity because Wave 1 of MIDUS did not ask respondents to report their ethnicity.

A composite measure of SES was based on education, occupation, income, and wealth, which we converted to a percentile rank representing the individual's position within the distribution at that survey wave (see S3 Text for details).

We also include selected chronic conditions (i.e., cancer, heart trouble, stroke, diabetes, lung problems) and physical limitations (see S4 Text for details).

## Analytic strategy

We used standard practices of multiple imputation to handle missing data [18, 19]. See S5 Text for details.

We fit Cox hazard models to estimate age-specific mortality, using age as the time metric. First, we tested daily exposure to stressors. In a second alternative model, we tested daily perceived stress. A robust variance estimator was used to correct for family-level clustering. In addition to age, all models controlled for sex, race, and marital status. In subsequent models, we adjusted sequentially for potential confounders of the association between stressors/severity and mortality: SES; chronic conditions; and physical function.

We tested the proportionality assumption for each of the covariates and found evidence that the hazard ratio (HR) varied significantly by age for the following covariates: race, married/partnered, socioeconomic status, exposure to stressors, severity of stress, and diabetes. Thus, the final models included interactions between age and those covariates.

In auxiliary analyses, we fit separate cause-specific models for extrinsic vs. other mortality. The first model included daily exposure to stressors, while the second model substituted daily perceived stress. All models were fully-adjusted for potential confounders.

All analyses were conducted using Stata 16.1 [20]. Statistical tests were two-sided at the 0.05 level.

## Results

Table 1 shows descriptive statistics by survey wave for all the covariates included in the analysis. The analytic sample covers a wide age range (21–95 at the NSDE, when exposure/severity of stress was measured), but most respondents were aged 26–75 at Wave 1 of the NSDE

(1996–97), aged 35–85 at Wave 2 (2004–09), and aged 47–93 at Wave 3 (2017–19). Among those who died by December 31, 2021, the youngest death occurred at age 30 and the oldest at age 97. The mean daily number of stressors was 0.5 (on average, respondents reported one stressor on half of observed days), with 12% of the analytic sample reporting no stressors and less than 2% reporting an average of 2–5 stressors per day. The mean daily perceived severity of stress was 1.0 (on average, "not very" stressful for one stressor across observed days), with 15% scoring zero and 13% scoring 2–14 per day.

Table 2 shows that the hazard ratio (HR) for daily exposure to stressors diminished with age. In the demographic-adjusted model (Model 1), the HR was 1.21 (95% CI 1.02–1.43) per SD of daily stressor exposure at age 50, but there was virtually no association by age 70 (HR = 1.01, 95% CI 0.92–1.11). We found a similar result when we substituted the measure of the daily perceived severity of stress (Table 3, Model 1): the HR was 1.23 (1.04–1.46) per SD at age 50, but declined to 1.06 (0.97–1.17) at 70.

Thus, we find little difference in the associations with exposure to stressors versus perceived severity of stress. However, exposure and severity are highly correlated (r = 0.93). Anyone who reported no stressors also scored zero on severity, while virtually everyone who reported an average of at least one stressor per day (17% of the sample) also scored above the 65th percentile of severity.

The associations strengthened a bit when we further adjusted for SES because SES was weakly, but positively correlated with daily stressors/severity. After controlling for SES, the HR at age 50 was 1.27 (1.08–1.50) per SD of exposure (Table 2, Model 2) and 1.26 (1.07–1.48) per SD of severity (Table 3, Model 2).

With further adjustment for chronic conditions and physical limitations, the relationship with exposure to daily stress weakened but remained significant at younger ages (HR = 1.20 per SD at age 50, 95% CI: 1.01–1.42, Table 2, Model 4); by age 70, the association was virtually nil (HR = 1.01, 95% CI: 0.92–1.11). The association with the alternative measure of severity was no longer significant even at age 50 (HR = 1.17 per SD, 95% CI: 0.99–1.37, Table 3, Model 4).

Table 4 compares the associations with extrinsic versus other mortality from the fully-adjusted model. The association between exposure to stressors (Model 1) and mortality at age 50 was somewhat stronger (and significant) for extrinsic mortality (HR = 1.31, 1.01–1.69) than for other mortality, which was not significant (HR = 1.24, 0.98–1.56). In contrast, the association with perceived severity (Model 2) and mortality at age 50 was not significant for either extrinsic (HR = 1.20, 0.92–1.58, Model 2) or other mortality (HR = 1.25, 0.99–1.57).

## Discussion

In demographic-adjusted models, daily exposure to stressors and perceived severity of stress were associated with increased mortality rates in midlife, but only at younger ages. The association was attenuated somewhat after controlling for chronic conditions and physical limitations, but the association between daily exposure to stressors and mortality in midlife remained notable and significant. The association with perceived severity was no longer significant.

Contrary to expectations, we found little difference between results for extrinsic versus intrinsic mortality. The association with exposure to stressors was only slightly stronger for extrinsic mortality than for other mortality, but perceived severity was not significantly associated with either type of mortality.

StrengthsAs noted in the Introduction, most prior studies of the association between perceived stress and mortality were based on one-time measurements of perceived stress [6–8, 11, 12]. Two other studies [9, 10] measured perceived stress at several follow-up waves, but

**Table 2. Hazard ratios for daily exposure to stressors from Cox models predicting age-specific mortality.**

| | Model 1 | Model 2 | Model 3 | Model 4 |
|---|---|---|---|---|
| Male | 1.25* | 1.30** | 1.20* | 1.33** |
| | (1.05–1.49) | (1.09–1.54) | (1.01–1.44) | (1.11–1.59) |
| Black | 2.51** | 1.96* | 1.62 | 1.50 |
| | (1.41–4.48) | (1.09–3.53) | (0.89–2.94) | (0.82–2.74) |
| Other races | 1.03 | 1.10 | 1.10 | 1.03 |
| | (0.23–4.59) | (0.25–4.88) | (0.25–4.90) | (0.23–4.67) |
| Married/partnered | 0.39*** | 0.46*** | 0.47*** | 0.48** |
| | (0.25–0.61) | (0.29–0.72) | (0.30–0.73) | (0.31–0.75) |
| SES percentile rank[a] | | 0.18*** | 0.27** | 0.38* |
| | | (0.08–0.41) | (0.12–0.61) | (0.17–0.86) |
| Cancer | | | 0.98 | 0.94 |
| | | | (0.80–1.21) | (0.76–1.17) |
| Heart trouble | | | 1.43*** | 1.27* |
| | | | (1.19–1.72) | (1.05–1.53) |
| Stroke | | | 1.46 | 1.31 |
| | | | (0.92–2.31) | (0.84–2.04) |
| Diabetes | | | 4.48*** | 3.74*** |
| | | | (2.77–7.25) | (2.28–6.13) |
| Lung problems | | | 1.29* | 1.13 |
| | | | (1.02–1.63) | (0.89–1.44) |
| Index of physical limitations[b] | | | | 1.42*** |
| | | | | (1.26–1.61) |
| Daily exposure to stressors[b] | 1.21* | 1.27** | 1.24* | 1.20* |
| | (1.02–1.43) | (1.08–1.50) | (1.05–1.46) | (1.01–1.42) |
| **Interactions with (Age-50):** | | | | |
| Black | 0.97** | 0.97* | 0.97* | 0.98 |
| | (0.94–0.99) | (0.95–0.99) | (0.95–1.00) | (0.95–1.00) |
| Other races | 0.96 | 0.96 | 0.96 | 0.96 |
| | (0.90–1.03) | (0.90–1.03) | (0.90–1.02) | (0.90–1.03) |
| Married/partnered | 1.03** | 1.02** | 1.02* | 1.02* |
| | (1.01–1.04) | (1.01–1.04) | (1.00–1.04) | (1.00–1.04) |
| SES percentile rank[a] | | 1.04** | 1.03* | 1.02 |
| | | (1.01–1.07) | (1.00–1.06) | (0.99–1.05) |
| Diabetes in the past 12 months | | | 0.97*** | 0.97** |
| | | | (0.95–0.99) | (0.96–0.99) |
| Daily exposure to stressors[b] | 0.991* | 0.991** | 0.991** | 0.992* |
| | (0.98–1.00) | (0.98–1.00) | (0.98–1.00) | (0.98–1.00) |

The 95% confidence intervals are shown in parentheses below the hazard ratio. In cases where there was evidence of non-proportional hazards, we interacted the relevant variable with Age-50 so that main effect represents the hazard ratio (HR) at age 50. For example, in Model 1, the HR for daily exposure to stressors at age 50 was 1.21. The corresponding HR for age $x$ can be obtained as follows: $HR^{Stressors} \times (HR^{Age \times Stressors})^{(x-50)}$, where $HR^{Stressors}$ is the HR for the main effect and $HR^{Age \times Stressors}$ is the HR for the interaction with age. Thus, the HR for stressors at age 70 is: $1.21*0.991^{20} = 1.01$.

[a] The HR represents the difference between the top (1) and bottom percentile (0) of SES.

[b] Standardized (based on the pooled distribution of observations from all waves); the HR represents the relative hazard per SD.

* $p<0.05$;

** $p<0.01$;

*** $p<0.001$

**Table 3. Hazard ratios for daily perceived severity of stress from Cox models predicting age-specific mortality.**

| | Model 1 | Model 2 | Model 3 | Model 4 |
|---|---|---|---|---|
| Male | 1.27** | 1.32** | 1.22* | 1.34** |
| | (1.06–1.51) | (1.10–1.57) | (1.02–1.46) | (1.12–1.61) |
| Black | 2.50** | 1.96* | 1.63 | 1.51 |
| | (1.40–4.46) | (1.10–3.51) | (0.90–2.94) | (0.83–2.73) |
| Other races | 1.07 | 1.13 | 1.13 | 1.04 |
| | (0.24–4.74) | (0.26–4.95) | (0.26–5.00) | (0.23–4.73) |
| Married/partnered | 0.39*** | 0.46*** | 0.47** | 0.49** |
| | (0.25–0.62) | (0.30–0.73) | (0.30–0.74) | (0.31–0.76) |
| SES percentile rank[a] | | 0.19*** | 0.29** | 0.40* |
| | | (0.08–0.43) | (0.13–0.64) | (0.18–0.89) |
| Cancer | | | 0.98 | 0.94 |
| | | | (0.80–1.21) | (0.76–1.16) |
| Heart trouble | | | 1.41*** | 1.25* |
| | | | (1.17–1.69) | (1.04–1.51) |
| Stroke | | | 1.47 | 1.32 |
| | | | (0.92–2.33) | (0.85–2.06) |
| Diabetes | | | 4.43*** | 3.72*** |
| | | | (2.74–7.16) | (2.27–6.10) |
| Lung problems | | | 1.28* | 1.13 |
| | | | (1.01–1.61) | (0.89–1.43) |
| Index of physical limitations[b] | | | | 1.42*** |
| | | | | (1.25–1.61) |
| Daily perceived severity of stress[b] | 1.23* | 1.26** | 1.21* | 1.17 |
| | (1.04–1.47) | (1.07–1.48) | (1.04–1.42) | (0.99–1.37) |
| **Interactions with (Age-50):** | | | | |
| Black | 0.97** | 0.97* | 0.97* | 0.98 |
| | (0.94–0.99) | (0.95–0.99) | (0.95–1.00) | (0.95–1.00) |
| Other races | 0.96 | 0.96 | 0.96 | 0.96 |
| | (0.90–1.03) | (0.90–1.03) | (0.89–1.02) | (0.90–1.03) |
| Married/partnered | 1.02** | 1.02* | 1.02* | 1.02* |
| | (1.01–1.04) | (1.00–1.04) | (1.00–1.04) | (1.00–1.04) |
| SES percentile rank[a] | | 1.04* | 1.03 | 1.02 |
| | | (1.01–1.07) | (1.00–1.06) | (0.99–1.05) |
| Diabetes in the past 12 months | | | 0.97*** | 0.97** |
| | | | (0.95–0.99) | (0.96–0.99) |
| Daily perceived severity of stress[b] | 0.9925* | 0.9929* | 0.9934* | 0.9943 |
| | (0.99–1.00) | (0.99–1.00) | (0.99–1.00) | (0.99–1.00) |

The 95% confidence intervals are shown in parentheses below the hazard ratio. In cases where there was evidence of non-proportional hazards, we interacted the relevant variable with Age-50 so that main effect represents the hazard ratio (HR) at age 50. For example, in Model 1, the HR for daily perceived severity of stress at age 50 was 1.23. The corresponding HR for age $x$ can be obtained as follows: $HR^{Severity} \times (HR^{Age \times Severity})^{(x-50)}$, where $HR^{Severity}$ is the HR for the main effect and $HR^{Age \times Severity}$ is the HR for the interaction with age. Thus, the HR for perceived severity of stress at age 70 is: $1.23*0.9925^{20} = 1.06$.

[a] The HR represents the difference between the top (1) and bottom percentile (0) of SES.

[b] Standardized (based on the pooled distribution of observations from all waves); the HR represents the relative hazard per SD.

* $p<0.05$;

** $p<0.01$;

*** $p<0.001$

**Table 4. Hazard ratios for daily stressors/severity from Cox models predicting age-specific extrinsic mortality[a] vs. other mortality,[b] fully-adjusted.**

| | Extrinsic Mortality | | Other Mortality | |
|---|---|---|---|---|
| | Model 1 | Model 2 | Model 1 | Model 2 |
| Male | 1.38 | 1.41 | 1.30* | 1.31* |
| | (0.95–1.99) | (0.97–2.04) | (1.05–1.61) | (1.06–1.62) |
| Black | 0.97 | 0.96 | 1.78 | 1.79 |
| | (0.28–3.34) | (0.29–3.12) | (0.82–3.87) | (0.83–3.86) |
| Other races | 1.12 | 1.08 | 0.99 | 1.01 |
| | (0.13–9.74) | (0.12–9.55) | (0.14–7.01) | (0.14–7.16) |
| Married/partnered | 0.61 | 0.63 | 0.45** | 0.45** |
| | (0.28–1.35) | (0.29–1.38) | (0.25–0.80) | (0.25–0.81) |
| SES percentile rank[c] | 0.24 | 0.26 | 0.49 | 0.51 |
| | (0.05–1.13) | (0.06–1.20) | (0.19–1.28) | (0.20–1.33) |
| Cancer | 1.03 | 1.02 | 0.91 | 0.91 |
| | (0.66–1.59) | (0.66–1.57) | (0.72–1.16) | (0.72–1.16) |
| Heart trouble | 1.26 | 1.24 | 1.24 | 1.23 |
| | (0.87–1.83) | (0.86–1.80) | (0.99–1.55) | (0.99–1.54) |
| Stroke | 0.90 | 0.89 | 1.66* | 1.65* |
| | (0.31–2.57) | (0.31–2.53) | (1.00–2.73) | (1.00–2.73) |
| Diabetes | 2.25 | 2.24 | 4.44*** | 4.45*** |
| | (0.89–5.67) | (0.90–5.56) | (2.30–8.55) | (2.32–8.56) |
| Lung problems | 2.02*** | 2.01*** | 0.85 | 0.85 |
| | (1.37–2.99) | (1.36–2.98) | (0.63–1.15) | (0.62–1.14) |
| Index of physical limitations[d] | 1.87*** | 1.86*** | 1.28*** | 1.27** |
| | (1.44–2.42) | (1.43–2.42) | (1.11–1.47) | (1.10–1.47) |
| Exposure to stressors[d] | 1.31* | | 1.24 | |
| | (1.01–1.69) | | (0.98–1.56) | |
| Perceived severity of stress[d] | | 1.20 | | 1.25 |
| | | (0.92–1.58) | | (0.99–1.57) |
| **Interactions with (Age-50):** | | | | |
| Black | 0.98 | 0.98 | 0.97 | 0.97 |
| | (0.93–1.03) | (0.93–1.03) | (0.95–1.00) | (0.95–1.00) |
| Other races | 0.91*** | 0.91*** | 0.97 | 0.97 |
| | (0.87–0.95) | (0.87–0.95) | (0.90–1.05) | (0.90–1.05) |
| Married/partnered | 1.01 | 1.00 | 1.03* | 1.03* |
| | (0.98–1.03) | (0.98–1.03) | (1.01–1.05) | (1.00–1.05) |
| SES percentile rank[c] | 1.04 | 1.03 | 1.01 | 1.01 |
| | (0.98–1.09) | (0.98–1.09) | (0.98–1.05) | (0.98–1.05) |
| Diabetes | 0.98 | 0.98 | 0.97* | 0.97** |
| | (0.94–1.01) | (0.94–1.01) | (0.95–0.99) | (0.95–0.99) |
| Exposure to stressors[d] | 0.9857* | | 0.9917 | |
| | (0.97–1.00) | | (0.98–1.00) | |

(*Continued*)

**Table 4.** (*Continued*)

| | Extrinsic Mortality | | Other Mortality | |
|---|---|---|---|---|
| | Model 1 | Model 2 | Model 1 | Model 2 |
| Perceived severity of stress[d] | | 0.9922 | | 0.9927 |
| | | (0.98–1.00) | | (0.98–1.00) |

The 95% confidence intervals are shown in parentheses below the hazard ratio. In cases where there was evidence of non-proportional hazards, we interacted the relevant variable with Age-50 so that main effect represents the hazard ratio (HR) at age 50. For example, the HR for exposure to stressors at age 50 was 1.31 for extrinsic mortality. The corresponding HR for age $x$ can be obtained as follows: $HR^{Stressors} \times (HR^{Age \times Stressors})^{(x-50)}$, where $HR^{Stressors}$ is the HR for the main effect and $HR^{Age \times Stressors}$ is the HR for the interaction with age. Thus, the HR for stressors at age 70 for extrinsic mortality is: $1.31*0.9857^{20} = 0.98$.

[a] Extrinsic mortality includes deaths from external causes, other alcohol/drug-related deaths, several causes that are strongly associated with smoking (e.g., COPD, lung cancer), and many infectious diseases.

[b] Other mortality includes all other ICD codes, most of which are among the leading causes of death (e.g., heart disease, other cancers, cerebrovascular disease, AD/dementia, diabetes) and are probably aging-related.

[c] The HR represents the difference between the top (1) and bottom percentile (0) of SES.

[d] Standardized (based on the pooled distribution of observations from all waves); the HR represents the relative hazard per SD.

* $p<0.05$;

** $p<0.01$;

*** $p<0.001$

measured it only once at each survey wave. In contrast, our measures of stress exposure and perceived severity come from a daily diary collected over an 8-day period. To our knowledge, there is no prior study of perceived stress and mortality that incorporated multiple measurements of perceived stress over a comparable period.

Our results for daily exposure to stressors were consistent with an earlier study [13], but we expanded on the earlier work to show that the association was stronger at younger ages, disappeared by age 70, and was somewhat stronger for extrinsic mortality than for mortality from other causes that were more likely to be aging-related. When we used an alternative measure of daily perceived severity of stress (not examined in [13]), the demographic-adjusted association appeared to be similar in magnitude, but after careful adjustment for potential confounding with health status, the association weakened and was no longer statistically significant. Finally, our analysis adjusted for a wide range of potential confounders that could influence both the perception of stress and risk of death. Our results suggest that the association between daily perceived severity of stress and mortality may be primarily a result of confounding with health status.

## Limitations

We purposefully did not include health behaviors (e.g., smoking, drug/alcohol abuse, physical activity) in our model because we suspect the relationship with perceived stress is reciprocal. People may be more susceptible to adverse health behaviors when they feel stressed, but adverse health behaviors probably exacerbate levels of perceived stress as well. It may be impossible to disentangle the direction of causality. If we controlled for health behaviors, we could not determine whether they represent confounders (i.e., they affect stress levels, which in turn influence mortality risk) or mediators (i.e., stress levels affect health behaviors, which in turn shape mortality risk).

Another limitation of this study is the lag (up to 15 years) between measurement of stress exposure/severity and the timing of death. The association with mortality might be stronger if we had shorter intervals between survey waves.

Like other observational studies, this study is subject to non-response bias. If individuals who experienced more stressors were less likely to participate in the survey and suffered higher mortality than participants, then our results may underestimate the association between stress and mortality. Among NSDE participants, only 62% completed the NSDE on all 8 days. All NSDE respondents were included in our analysis, even if they were not observed on all 8 days. However, there was an inverse correlation ($r$ = -0.15) between the stress measures (exposure & severity) and the number of observed days. For example, the average daily number of stressors was 0.5 among respondents observed for 8 days, whereas it was 0.8 for those observed for 4 or fewer days. If respondents were less likely to complete the NSDE on high stress days, we would underestimate the stress burden in the population, which could bias the estimated association with mortality.

## Implications

A key implication for future research on this topic is the importance of carefully controlling for health status as a confounder and evaluating the extent to which the relationship varies by age and cause-of-death. For example, the association with stress exposure/severity may be stronger for mortality from external causes—and deaths of despair (from drug/alcohol abuse and suicide) more generally—than for other extrinsic deaths, most of which were probably related to smoking. Smoking generally takes a long time to kill people and those deaths generally occur at older ages, whereas deaths from external causes and other deaths of despair commonly occur at young to middle ages. We were unable to model cause-specific mortality in more detail because of the limited number of deaths in our analytic sample.

For clinicians, concern about stress exposure may be better focused on patients in midlife, who may be exposed to higher levels of stress related to work and who may be more susceptible to extrinsic mortality, particularly deaths related to drug and alcohol use, suicide, and other external causes. In our analytic sample, those who died below age 60 were more likely to die of extrinsic causes (41%) than those who died at older ages (24% of deaths at ages 60–89 and 18% of deaths at ages 90+). In particular, those dying from drug/alcohol use, suicide, or other external causes comprised 24% of deaths below age 60 but only 5% or less of deaths at older ages.

Clinicians should also be aware that subjective measures of perceived stress are susceptible to endogeneity (i.e., underlying health conditions may influence a person's perception of stress). Attempts to ameliorate perceived stress without addressing the underlying health conditions that could have generated the perception of stress may be fruitless. Perceived stress may be a consequence rather than a cause of heightened mortality risk. To effectively reduce mortality risk, we need to target the root causes of perceived stress.

It may be more useful for clinicians to ask about exposure to stressors (i.e., objective questions about the extent to which a person experienced particular events that are stressful for most people) rather than perceived stress (i.e., subjective questions such as "How stressful was this event for you?"). For individuals exposed to stressors, clinicians could encourage practices that facilitate better coping with stress (e.g., relaxation techniques such as mediation, yoga, or deep breathing, physical activity, better quality sleep, avoiding excessive use of alcohol, tobacco, or other substances) [21, 22].

## Conclusion

Daily exposure to stressors is associated with mortality only at younger ages, and the association is slightly stronger for extrinsic than other mortality. In contrast, the association between daily perceived severity of stress and mortality appears to be largely a result of confounding

with health status. These results suggest that perceived stress is more susceptible to endogeneity (i.e., ill health influences both the perception of stress and mortality) than stress exposure. Most Americans die at older ages, where stress exposure does not appear to be significantly associated with mortality. Stress exposure is more strongly associated with midlife mortality, which has an undue influence on overall life expectancy. According to Nietzsche [23], "What doesn't kill me makes me stronger." We need more young Americans to survive.

## Supporting information

**S1 Text. MIDUS main survey and NSDE.**
(PDF)

**S2 Text. Mortality follow-up.**
(PDF)

**S3 Text. Measure of socioeconomic status.**
(PDF)

**S4 Text. Measures of health status.**
(PDF)

**S5 Text. Multiple imputation.**
(PDF)

**S1 File. Zip archive of the Stata do-files (i.e., author-generated code) used for this analysis.**
(ZIP)

## Author Contributions

**Conceptualization:** Dana A. Glei, Maxine Weinstein.

**Formal analysis:** Dana A. Glei.

**Funding acquisition:** Maxine Weinstein.

**Methodology:** Dana A. Glei.

**Project administration:** Maxine Weinstein.

**Software:** Dana A. Glei.

**Supervision:** Maxine Weinstein.

**Visualization:** Dana A. Glei.

**Writing – original draft:** Dana A. Glei.

**Writing – review & editing:** Dana A. Glei, Maxine Weinstein.

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
