## [Decision Letter · Decision Letter 0]

13 Mar 2024

PONE-D-24-02083Daily exposure to stressors, daily perceived severity of stress, and mortality risk among US adultsPLOS ONE

Dear Dr. Glei,

Thank you for submitting your manuscript to PLOS ONE. After careful consideration, we feel that it has merit but does not fully meet PLOS ONE’s publication criteria as it currently stands. Therefore, we invite you to submit a revised version of the manuscript that addresses the points raised during the review process.

We look forward to receiving your revised manuscript.

Kind regards,

Kallol Kumar Bhattacharyya, MBBS MA PhD

Academic Editor

PLOS ONE

Journal Requirements:

**Additional Editor Comments:**

You can see below that the reviewers have suggested some minor changes in the manuscript to make it a stronger paper; therefore, I invite you to make those changes and resubmit the manuscript. You are very close.

Reviewers' comments:

Reviewer's Responses to Questions

**Comments to the Author**

1. Is the manuscript technically sound, and do the data support the conclusions?

Reviewer #1: Yes

Reviewer #2: Yes

2. Has the statistical analysis been performed appropriately and rigorously? 

Reviewer #1: Yes

Reviewer #2: Yes

3. Have the authors made all data underlying the findings in their manuscript fully available?

Reviewer #1: Yes

Reviewer #2: Yes

4. Is the manuscript presented in an intelligible fashion and written in standard English?

Reviewer #1: Yes

Reviewer #2: Yes

5. Review Comments to the Author

Reviewer #1: 1. This paper is well done.

2. Just thinking broadly, could the findings be different among women, white or Hispanic non-white populations? The introductory part could incorporate more information by comparing gender, race among other demographic variables. Older adult population is very diverse and doing a comparative analysis could reveal insights for future policy, research and practice implications.

3. It will be great to include the practice implications of this paper and how it can improve the health and well-being of older adults.

4. Considering the increase in stress levels among younger adults as the study findings indicate, how does this translate to better health outcomes as they age across the lifespan and into older adult years. Are there any implications or recommendations that could positively shape stress and mortality rates among the U.S. population?

Reviewer #2: Dear Editor,

Thank you for the opportunity to review the manuscript, "Daily exposure to stressors, daily perceived severity of stress, and mortality risk among US adults."

Overall, I thought the authors did a good job explaining the study. The manuscript started with quotes relevant to the study and thought it was a good set up to discuss exposure to stressors and daily perceived severity of stress on mortality risk. I appreciated the supportive files as they help me understand the rationale and reasoning of the authors and study. The tables were excellent. I thought the authors did a great job on constructing the tables with explanations (notes) at the bottom.

I have a few things for the authors to address. In the 2nd paragraph of the intro, "i.e., the extent to which s/he feels..." should include gender neutral language. Can be done by changed the statement to, she/he/they. Same goes for throughout the manuscript. In the 3rd paragraph of the intro, "Most studies are based on one-time..." could use citations to let the reader know which studies the authors are referring to. In the 2nd paragraph of the measures, I think there might be a typo: analysis should be analytical? In the 4th paragraph of the measures section, perhaps add a citation to be consistent with the previous paragraph. In the Daily exposures to stressors section, the authors did not include ranges, maybe be further elaborate. In the Daily perceived severity of stressors sections in hard to follow, maybe further elaborate. In the discussion, the manuscript would improve if the authors clearly labeled the limitations, but also included the strengths of the study. The authors could expand on the conclusion by including possible implications perhaps.

I am unfamiliar with the guidelines for authors regarding word counts, etc. Some of these comments may not be appliable.

I recommend the manuscript be accepted with minor revisions.

6. PLOS authors have the option to publish the peer review history of their article (what does this mean?). If published, this will include your full peer review and any attached files.

Reviewer #1: No

Reviewer #2: No

---

## [Author Response · Author response to Decision Letter 0]

1 Apr 2024

See our (attached) letter responding to the reviewers comments.

---

## [Decision Letter · Decision Letter 1]

23 Apr 2024

Daily exposure to stressors, daily perceived severity of stress, and mortality risk among US adults

PONE-D-24-02083R1

Dear Dr. Glei,

We’re pleased to inform you that your manuscript has been judged scientifically suitable for publication and will be formally accepted for publication once it meets all outstanding technical requirements.

Kind regards,

Kallol Kumar Bhattacharyya, MBBS MA PhD

Academic Editor

PLOS ONE

Additional Editor Comments (optional):

Reviewers' comments:

Reviewer's Responses to Questions

**Comments to the Author**

1. If the authors have adequately addressed your comments raised in a previous round of review and you feel that this manuscript is now acceptable for publication, you may indicate that here to bypass the “Comments to the Author” section, enter your conflict of interest statement in the “Confidential to Editor” section, and submit your "Accept" recommendation.

Reviewer #1: All comments have been addressed

Reviewer #2: All comments have been addressed

2. Is the manuscript technically sound, and do the data support the conclusions?

Reviewer #1: Yes

Reviewer #2: Yes

3. Has the statistical analysis been performed appropriately and rigorously? 

Reviewer #1: Yes

Reviewer #2: Yes

4. Have the authors made all data underlying the findings in their manuscript fully available?

Reviewer #1: Yes

Reviewer #2: Yes

5. Is the manuscript presented in an intelligible fashion and written in standard English?

Reviewer #1: Yes

Reviewer #2: Yes

6. Review Comments to the Author

Reviewer #1: This manuscript looks good for publication. The authors provide insightful information that are beneficial to readers and target population (US adults).

Reviewer #2: Thank you for the opportunity to review this revised manuscript. I believe it is ready for acceptance. The authors addressed reviewer comments.

7. PLOS authors have the option to publish the peer review history of their article (what does this mean?). If published, this will include your full peer review and any attached files.

Reviewer #1: No

Reviewer #2: No

---

## [Editor Report · Acceptance letter]

3 May 2024

PONE-D-24-02083R1 

PLOS ONE

Dear Dr. Glei, 

I'm pleased to inform you that your manuscript has been deemed suitable for publication in PLOS ONE. Congratulations! Your manuscript is now being handed over to our production team.

Kind regards, 

on behalf of

Dr. Kallol Kumar Bhattacharyya 

Academic Editor

PLOS ONE